# The Role of the Epididymis and the Contribution of Epididymosomes to Mammalian Reproduction

**DOI:** 10.3390/ijms21155377

**Published:** 2020-07-29

**Authors:** Emma R. James, Douglas T. Carrell, Kenneth I. Aston, Timothy G. Jenkins, Marc Yeste, Albert Salas-Huetos

**Affiliations:** 1Andrology and IVF Laboratory, Division of Urology, Department of Surgery, University of Utah School of Medicine, Salt Lake City, UT 84108, USA; emma.james@utah.edu (E.R.J.); douglas.carrell@hsc.utah.edu (D.T.C.); ki.aston@hsc.utah.edu (K.I.A.); 2Department of Human Genetics, University of Utah School of Medicine, Salt Lake City, UT 84112, USA; 3Department of Physiology and Developmental Biology, Brigham Young University, Provo, UT 84604, USA; tim_jenkins@byu.edu; 4Biotechnology of Animal and Human Reproduction (TechnoSperm), Unit of Cell Biology, Department of Biology, Faculty of Sciences, Institute of Food and Agricultural Technology, University of Girona, 17003 Girona, Spain; marc.yeste@udg.edu

**Keywords:** exosomes, epididymis, epididymosomes, reproduction, spermatozoa, sperm transport, sperm maturation

## Abstract

It is well-established that testicular spermatozoa are immature and acquire motility and fertilization capabilities during transit throughout the epididymis. The epididymis is a duct-like organ that connects the testis to the vas deferens and is comprised of four anatomical regions: the initial segment, caput, corpus, and cauda. Sperm maturation occurs during epididymal transit by the interaction of sperm cells with the unique luminal environment of each epididymal region. In this review we discuss the epididymis as an essential reproductive organ responsible for sperm concentration, maturation (including sperm motility acquisition and fertilizing ability), protection and storage. Importantly, we also discuss specific characteristics and roles of epididymal-derived exosomes (epididymosomes) in establishing sperm competency within the intricate process of reproduction. This review suggests that an increasing body of evidence is working to develop a complete picture of the role of the epididymis in male reproduction, offspring health, and disease susceptibility.

## 1. Introduction

The number of published studies with the MeSH term “epididymis” (>13,000) has been growing year by year. Within the last two years (2018-19), more than 300 articles including this term have been published according to PubMed-MEDLINE database (https://pubmed.ncbi.nlm.nih.gov/). This evidence documents an increased interest in this important reproductive organ in recent years and suggests emerging research which could add to current knowledge of reproductive biology.

It is well known that testicular spermatozoa are immature and only acquire motility and fertilizing ability during transit through the epididymis [1]. This organ consists of a long, convoluted tubule connecting the efferent ducts of the testis to the vas deferens. The epididymis has four main anatomical regions – the initial segment, caput, corpus and cauda – each with unique characteristics and functions [2]. During transit of spermatozoa through the epididymis, a wide variety of changes occur within the epididymal lumen environment. These changes include the release and absorption of fluids, ions, antioxidants, and of particular importance to this review, exosomes known as “epididymosomes” [3,4,5,6].

In this review we discuss the process of mammalian reproduction and specifically the role of the epididymis as an essential reproductive organ. Many well described roles of the epididymis are reviewed, such as sperm concentration, protection, transport and storage, among others. Additionally, we discuss evidence for newly hypothesized roles of the epididymis related to epididymosomes and explore existing basic science research that impacts sperm competency and cell-to-cell communication within the male reproductive tract.

## 2. The Reproductive Process

Reproduction is defined as “the natural process among organisms by which new individuals are generated and the species perpetuated” [1]. In mammals, the reproductive process is sexual, fertilization usually occurs internally, and the fertilized oocyte is developed in the uterus. The placentals (or Eutheria order), with nearly 4000 described species, have some differences in terms of embryo and fetal development as compared to Monotremata and Metatheria [7,8]. In this review, we are going to briefly discuss pertinent aspects of the reproductive process of placental mammals, to provide a reproductive context for the basic science principles that will be discussed further.

In reproduction, two preliminary processes are essential to successfully generate a new organism: spermatogenesis and oogenesis. In male and female embryos, a number of embryonic cells in the epiblast enter the germ cell lineage to become primordial germ cells (PGC), the stem cells or building blocks of gametogenesis [9].

### 2.1. Spermatogenesis: From PGC to Spermatozoa

The creation of a male gamete (also known as spermatozoa or sperm) begins with the differentiation of primordial germ cells (PGC) into spermatogonial stem cells. Spermatogenesis then begins within the seminiferous tubules of the testis. Spermatogenesis initiates at the basal membrane, at the outermost portion of the Sertoli cells that line the seminiferous tubules, and progresses towards the tubule lumen. The concentration of retinoic acid (vitamin A) along the seminiferous tubule is an essential factor influencing the activation and support of spermatogenesis [10,11]. Undifferentiated “A” spermatogonia develop into differentiated “B” spermatogonia through a series of specialized mitotic divisions. A final mitotic division results in the formation of pre-leptotene primary spermatocytes. This step is often considered the point of entry into meiosis. Primary spermatocytes then undergo meiosis I to produce two secondary spermatocytes. Finally, each secondary spermatocyte divides into two equal haploid, round spermatids during meiosis II. These round spermatids undergo condensation and elongation to become elongated spermatids during a process called spermiogenesis [10]. When these cells are finally released from the Sertoli cells into the seminiferous tubule lumen they are considered immature spermatozoa and are ready to transit through the male reproductive tract and gain competency to fertilize an oocyte and support embryonic development.

### 2.2. Ogenesis: From PGC to Mature Ocytes

Oogenesis, in contrast to spermatogenesis, is the creation of a female gamete (also known as oocyte) from a PGC. In addition to generating a haploid gamete, the process of oogenesis must also generate many of the enzymes, mRNAs and other materials necessary to maintain preimplantation embryonic development [12]. This process involves three key phases: migration/proliferation, growth, and maturation. Oogenesis involves a progression from PCGs to primary oocytes, secondary oocytes, and finally to mature oocytes. Initiation of oogenesis begins with the migration of PCGs from the extra-embryonic mesoderm to the genital ridge. During this migration PCGs proliferate by undergoing mitosis to create a pool of oogonia. Over the growth phase, oogonia enter meiosis I and arrest in prophase I resulting in primary oocytes [12]. Simultaneous with the initiation of meiosis in human ovaries, germ cell cysts break, and pregranulosa cells surround the primary oocyte to form a single-layer primordial follicle. Primordial follicles present at birth determine the reservoir of germ cells available during the female reproductive lifespan [13]. The maturation phase begins after puberty when primary oocytes re-enter meiosis I and become secondary oocytes. Oocyte maturation occurs simultaneously with folliculogenesis under the influence of hormonal regulation [14]. Secondary oocytes are arrested in metaphase of meiosis II and are then considered mature oocytes which will either be ovulated or undergo atresia (degeneration). Once an ovulated oocyte has reached the ampulla of the fallopian tube it is competent to support embryonic development and, if fertilized by sperm, it will then reinitiate metaphase-II and complete meiosis II [15].

### 2.3. Transit of Gametes and the Fertilization Process

The path of an oocyte from the location of ovulation at the ovary to the location of fertilization in the fallopian tube, while highly regulated, is relatively short in distance [16]. The path that spermatozoa travel, however, is much longer. Spermatozoa exit the seminiferous tubules, travel through the efferent ducts and the length of the epididymis at the end of which they are stored prior to ejaculation [2]. Following ejaculation sperm travel through the vas deferens and then are deposited into the female reproductive tract where they must travel through the cervix and uterus to reach the site of fertilization, the ampulla of the fallopian tube [17]. In the female reproductive tract, sperm undergo capacitation, a final process necessary for the development of sperm competency. Capacitation results in many changes to spermatozoa, including hyper-motility, activation of some signaling pathways, and importantly, destabilization of the acrosomal region of the sperm head resulting in acrosome reaction and in increased capacity for fusion of the sperm to the egg [18]. At the site of fertilization, sperm must then bind and penetrate the oocyte vestments (corona radiata/cumulus cells and zona pellucida) for successful fertilization. Due to the extended journey that sperm undertake, highly motile cells are best poised to successfully reach the site of fertilization. Additionally, both oocytes and sperm must possess specific factors which allow the guiding of sperm to an oocyte, as well as appropriate binding and penetration by spermatozoa [19]. Motility as well as many of these specialized factors are acquired by spermatozoa during epididymal transit and largely via contact with epididymosomes [19,20].

## 3. The Epididymis: An Essential Reproductive Organ

The epididymis is one of the male sex accessory ducts that also include: the seminiferous tubules, rete testis, efferent ducts, vas deferens, ejaculatory duct, and urethra. The epididymis is a convoluted, crescent-shaped structure, about 3.8 cm long in humans, and is conserved in all male reptiles, birds, and mammals [1]. This organ consists of a long tubule that connects the testis to the vas deferens and has four main anatomical regions each with unique characteristics and functions: the initial segment, caput (head), corpus (body) and cauda (tail) (Figure 1) [2].

### 3.1. Epididymal Structure

To appreciate all of the functions of the epididymis, it is important to understand the structure of the tubules that make up this organ. There are six main cell types that make up the epididymal epithelium, some of these cells are found within all regions of the epididymis while others are localized to specific regions [2]. In general, epididymal cells have high metabolic, endocytic and secretory activity that is primarily regulated by androgens. Androgens are also responsible for regulating the synthesis of some, but not all, proteins that are synthesized and secreted by epididymal cells [21]. Principal cells are the major cell type in the epididymal epithelium and exist along the entire epididymal duct. Depending on the region, principal cells account for between 65 and 80% of the epididymal epithelium. These cells are primarily responsible for absorption and secretion of materials into the epididymal lumen and therefore have high secretory and endocytic activity [2,22]. Additionally, principal cells are the site of production and release of cargo-containing epididymosomes [5]. Apical cells are primarily located at the initial segment of the epididymal epithelium and also have endocytic activity. Narrow cells also exist exclusively within the initial segment and are, as the name indicates, narrower than adjacent principal cells. These cells have been shown to secrete H^+^ ions into the epididymal lumen, and are responsible for endocytosis [22]. Clear cells are another cell type with high endocytic activity; however, these cells are found exclusively within the caput, corpus and cauda regions of the epididymis and are not located within the initial segment. Clear cells are the primary cell type responsible for taking up cytoplasmic droplets that are released from sperm cells during maturation in the epididymal lumen. Together, clear and narrow cells are thought to be the primary cells responsible for the regulation of luminal pH [22]. Basal cells are located along the tubule and adhere to the basement membrane [2]. These cells are an integral part of tubule structure and it has been suggested that they may indirectly affect luminal environment by regulating some principal cell functions [2,22,23]. Finally, halo cells exist throughout the epididymal epithelium and are the primary immune cells in the epididymis [2]. The epididymal epithelium is also surrounded by smooth muscle which is thinnest at the caput and gets progressively thicker towards the cauda epididymis. In fact, the cauda is surrounded by two unique smooth muscle layers while the caput is encapsulated by a single layer [24].

### 3.2. Epididymal Functions

#### 3.2.1. Sperm Transport

The most obvious function of the epididymis is to transport sperm from the rete testes to the vas deferens. Total transit time through the epididymis is generally between 10–15 days [2]. Transport is achieved primarily by rhythmic contractions of the smooth muscle layers surrounding the epididymis. While contractions occur most frequently at the caput, they are most amplified at the cauda. Additionally, it has been suggested that cilia on epididymal epithelial cells may aid in directing sperm transit through the epididymis [24].

#### 3.2.2. Sperm Concentration

The main process that occurs in the initial segment of the epididymis is the absorption of fluid by epithelial cells. The efferent ducts and initial segment are responsible for absorbing approximately 90% of the fluid that leaves the rete testis [2]. Additional absorption occurs throughout the remainder of epididymal transit, resulting in a dramatic increase in sperm concentration in the cauda epididymis as compared to the rete testis [25]. Sperm concentration in the epididymis is necessary for increased sperm concentration in semen, an important factor in male fertility.

#### 3.2.3. Sperm Protection

An additional function of the epididymis is to protect sperm cells during epididymal transit from damage caused by the external environment [26]. The epididymis has many mechanisms to aid in the protection of sperm. Epididymal epithelial cells have high metabolic activity which results in the generation of reactive oxygen species that are harmful to sperm cells. To combat this problem, epithelial cells excrete various antioxidant enzymes, including superoxide dismutase, into the epididymal lumen to neutralize reactive oxygen species [27]. Additionally, a blood-epididymis barrier functions to shield maturing sperm cells from the immune system and from harmful substances that might exist in the bloodstream [2].

#### 3.2.4. Sperm Storage

The cauda epididymis functions as a storage location for functionally mature sperm cells prior to ejaculation [22]. At a given time, between 50 and 80% of sperm in the epididymal lumen are located in the cauda epididymis, depending on species [2]. Epithelial cells of the cauda secrete factors that help to maintain a luminal environment designed to maintain sperm in a quiescent state during storage. While many factors relating to this quiescent state are still unknown, regulation of luminal pH and the presence of specific proteins and enzymes are thought to play a role [2]. After ejaculation, sperm leave this quiescent state, and metabolic activity increases 3–5 fold as compared to activity in the cauda epididymis [28].

#### 3.2.5. Sperm Maturation

An essential process required for normal male fertility that occurs during epididymal transit is sperm maturation. Sperm undergo many maturational changes during this time, but most importantly they acquire motility and factors necessary for successful fertilization of an oocyte. The process of maturation occurs via direct contact of sperm with the contents of the epididymal lumen environment. Luminal environment is specific to each region of the epididymis and differences between regions are due to the varied cell composition of the epithelium and hormonal regulation, among other factors [22]. As sperm progress through the epididymis, they undergo changes in nuclear compaction, plasma membrane composition, cytoskeletal structure, protein payload and non-coding RNA payload [22,29].

#### 3.2.6. Acquisition of Motility

Testicular sperm are considered immotile. While these cells may twitch, they are unable to complete any progressive movement. By the time spermatozoa reach the cauda epididymis a majority of cells are capable of progressive motility. Motility is thought to be intrinsic to sperm cells and can be artificially developed in certain conditions [20]. The epididymal lumen, however, provides the best environment for this activation to take place. From a morphological and structural perspective, many changes to sperm occur which help to facilitate motility [2]. Sperm plasma membrane composition is altered throughout epididymal transit resulting in a narrowing of the sperm acrosome. Alterations in membrane composition are thought to be driven by concentration gradients of specific enzymes and molecules along the tubule lumen [20,22]. Increased numbers of disulfide bridges in the sperm nucleus result in compaction of genetic material and the sperm head. Additionally, the cytoplasmic droplet, which is eventually shed upon ejaculation, migrates caudally along sperm during epididymal transit, and this droplet has been implicated in affecting some biochemical aspects of motility. Importantly, some signaling pathways have been associated with sperm motility, and evidence suggests that sperm may have functional flagellar machinery that is activated during epididymal transit [20].

#### 3.2.7. Fertilization Capabilities

In addition to the development of motility, sperm also gain factors necessary for fertilization of an oocyte during maturation in the epididymis. Specifically, they acquire factors necessary for binding and penetrating the cumulus cells and zona pellucida. It has been well established that the primary mechanism responsible for sperm-oocyte binding is carbohydrate-protein interactions between oligosaccharides on the oocyte membrane and receptor proteins on the sperm membrane [30]. The sperm acrosome reaction is required for sperm penetration and takes place within the fallopian tube. Many factors work in concert to initiate and complete the acrosome reaction, and oocyte-sperm carbohydrate-protein interactions are a key factor in this process [18]. Multiple proteins on the sperm surface are important for sperm-zona binding, including ZP proteins, acrosin binding protein and CRISP1, among others [18,19,31,32]. Binder of SPerm (BSP) family proteins also play a role in this process by stabilizing the sperm membrane and regulating timing of sperm capacitation [33]. Multiple ADAM family proteins are also important in sperm-zona binding, and while many of these proteins are expressed in the testis, some studies have shown increased levels of these proteins in sperm after epididymal transit [34,35,36]. Studies assessing the sperm proteome throughout epididymal transit have shown that these proteins, among others, are acquired by maturing sperm during epididymal transit [20,22,32,34,35,37,38]. These data, as well as early works assessing the fertilization capabilities of caput sperm for in vitro fertilization provide evidence for the epididymis having an essential role in sperm competency.

### 3.3. Sperm Maturation and Reproduction

New and historical data point to the importance of proper sperm maturation in successful reproduction. In fact, evidence suggests that the consequence of immature sperm on reproductive potential may go beyond a reduced ability to successfully fertilize an oocyte and extend into the support of early embryogenesis.

Historically, gross defects in embryogenesis from caput-sperm-fertilized oocytes have been well documented [2,39,40,41,42]. In the past, in vitro fertilization (IVF) was used in animal models as a method to study the effects of using immature (caput) sperm for reproduction. Due to these cells not having undergone complete epididymal transit, they lacked many of the necessary fertilization factors, and therefore fertilization rates in these studies were very low. In those embryos where fertilization did occur, however, gross defects in embryonic development were observed. Early studies in multiple model systems have suggested that sperm acquire fertilization factors relatively early during epididymal transit, but that the ability to produce viable offspring and full litters is not gained until later in the epididymis [2,39,40,41,42]. In 1977, a study was conducted where rabbit oocytes were fertilized with either immature corpus sperm or cauda sperm. These authors found that in corpus-derived embryos the first cleavage was consistently delayed as compared to cauda-derived embryos [41]. Likewise, in 1990, caput and cauda sperm were compared in an IVF study using mice. In successfully fertilized caput embryos, only approximately 8% developed to the blastocyst stage compared to approximately 48% of cauda-derived embryos [42].

Developments in assisted reproductive technologies have since allowed additional exploration into these observations, specifically with the use of intracytoplasmic sperm injection (ICSI) which does not require sperm to be motile, nor possess fertilization factors. A recent study by Conine et al. displayed striking results where ICSI embryos fertilized with caput sperm displayed significantly reduced implantation rates and ultimately 100% embryonic lethality [43]. These results were attributed to non-coding, small RNA (sRNA) payloads in caput sperm, as these payloads undergo significant remodeling during epididymal transit [43,44,45]. However, some data from other groups are in disagreement with these results, where authors successfully generated ICSI offspring using caput sperm [46,47]. Even considering inconsistencies in results, both historical and recent data suggest sperm competency to support normal embryogenesis may be acquired during epididymal transit. Recently, it has been suggested that cargo contained in epididymosomes and delivered to sperm during epididymal transit may be responsible for the acquisition of these critical factors [3,43,45].

## 4. Epididymosomes

Epididymosomes are exosomes that are found within the epididymal lumen and are released by the epididymal epithelium [5]. Exosomes are a distinct subtype of extracellular vesicles, along with apoptotic bodies and micro-vesicles. These subtypes are differentiated based upon their biogenesis, release pathways, content, size and function [48] Specifically, exosomes are produced in the endosomal compartment of most eukaryotic cells and were first identified in 1983 and termed ‘exosomes’ in 1987. Exosomes are secreted by most cell types and have been identified in plasma, urine, saliva, and semen, among other extracellular fluids [49]. Epididymosome is the specific term that is used to refer to exosomes within the epididymis.

As previously discussed, the epithelial cells of the epididymis are responsible for a multitude of secretions into the epididymal lumen in order to nourish and protect sperm during transit. Apocrine secretion from the principal cells of the epididymal epithelium results in a protruding apical bleb into the epididymal lumen. This bleb detaches and subsequently dissolves and releases cargo-containing epididymosomes into the lumen (Figure 2) [5]. These epididymosomes are then free to interact with other cells, including sperm cells and non-adjacent cells of the epithelium. Most exosomes, including epididymosomes have similar membrane lipid compositions and use similar sorting and docking mechanisms to aid in their role of cell-to-cell communication [50]. Epididymosomes specifically have essential roles in sperm competency and mounting evidence suggests they may be part of a mechanism for epigenetic inheritance of paternal traits and communication between cells of the epididymal epithelium.

### 4.1. Epididymosomal Proteins

Epididymosomes have been shown to contain hundreds of different proteins from various functional classes. Additionally, the protein composition of epididymosomes varies between different regions of the epididymis. One study examining proteomics of epididymosomes isolated from the bull epididymis revealed 555 proteins in caput-derived epididymosomes and 438 in cauda-derived epididymosomes with only 231 proteins identified in both subsets. Gene ontology analysis revealed associated functions for the identified proteins including as enzymes, transporters, structural proteins and chaperones, among others [51]. Additionally, a more recent study profiled proteins contained in mouse epididymosomes and identified a total of 1640 proteins in epididymosomes derived from the caput, corpus and cauda epididymis. Gene ontology analysis of these proteins suggested associated biological processes such as protein transport, oxidation-reduction processes, metabolic processes and others [52]. Many of the proteins identified in these data in the mouse have been previously identified as exosomal cargo, suggesting that not all epididymosomal cargo has specific or unique roles in reproduction. After removal of previously identified exosomal proteins from their analysis, however, Nixon et al. found that gene ontology analysis resulted in associations with biological processes such as spermatogenesis, sperm-zona pellucida binding and fertilization [52]. Interestingly, the synthesis of some proteins identified in rat epididymosomes such as methylmalonate-semialdehyde dehydrogenase (MMSDH) has been shown to be regulated by androgen levels [53]. This suggests that androgens may indirectly play a role in the establishment of epididymosomal cargo.

#### Functions of Epididymosomal Proteins

Many of the proteins previously discussed which are gained during epididymal transit and required for sperm fertilization capabilities have been identified in epididymosomes. Specifically, ZP family proteins ZP3R and ZPBP2 have been identified in mouse epididymosomes and are important for sperm-zona binding in the mouse [52]. Additionally, some ADAM family proteins have been observed in epididymosomes, including ADAM28, ADAM1, ADAM7 and conflicting reports of ADAM3 [51,52]. In a study of epididymosomes from the bull cauda epididymis, P25b and MIF were observed at a high abundance [38,51,54]. P25b is a protein which has been associated with subfertility in the bull [38,54]. MIF has been shown to play an important role in the development of sperm motility [55]. Direct transfer of protein cargo from epididymosomes to sperm has been confirmed in multiple studies [5,37,53]. Taken together with the characterization of epididymosomal cargo, this suggests that epididymosome-mediated transfer of proteins from the epididymal epithelium to maturing sperm cells is a primary mechanism for the establishment of sperm competency. Despite this, little is known regarding the mechanisms that regulate protein sorting into epididymosomes, the release of epididymosomes into the lumen, and the transfer of cargo from epididymosomes to the developing sperm. Considering the importance of epididymosomal protein cargo in sperm competency, additional research into these mechanisms and physiologic, molecular, or environmental conditions that may disrupt them is an important endeavor if this research is to be applied in clinical practice. Hormones are a primary regulator of epididymal cell functions, including protein synthesis and secretory actions, therefore, further research into interplay between hormones and epididymosomes could be an informative avenue.

### 4.2. Epididymosomal ncRNAs

A relatively recent addition to the knowledge base surrounding epididymosomes is the profiling of non-coding RNA species in epididymosome cargo. In general, ncRNAs in male reproduction has been a popular and somewhat controversial area of research in recent years. A vast repertoire of ncRNAs, specifically small RNAs (sRNAs) has been identified in epididymosomes, and multiple functions of these sRNAs have been proposed [6]. Small RNAs are a subset of ncRNAs that include primarily microRNAs (miRNAs), piwiRNAs (piRNAs), ribosomal small RNAs (rsRNAs) and tRNA derived small RNAs (tsRNAs) (also referred to as tRNA fragments or tRFs) [56]. Small RNA species such as miRNAs are capable of inhibiting gene translation by targeting mRNAs for degradation [57]. The abundance of sRNA species and the relative proportions of specific species such as miRNAs, piRNAs and tsRNAs in sperm has been found to vary between epididymal regions [44,45]. To determine whether epididymosomes may mediate these changes, sRNA content of epididymosomes has been interrogated. MicroRNAs and tsRNAs are the most abundant sRNAs in epididymosomal cargo, and many potential reproductive roles have been proposed for these sRNA species [45].

#### 4.2.1. Micro RNAs

Early reports using microarrays to assess miRNA content in epididymosomes presented evidence for a high abundance of miRNAs in epididymosomes. A study in bull epididymosomes identified 1645 miRNA sequences with miRNA species from the let-7 and miR-200 families as well as miR-26a, miR-103 and miR-191 being most highly represented. These authors also determined that, like epididymosome protein cargo, miRNA cargo differs significantly between segments of the epididymis [58]. A next generation sequencing study in mouse epididymosomes revealed complementary results, identifying 358 miRNAs within epididymosomes. This group found that approximately 68% of miRNAs existed across three segments of the epididymis (caput, corpus and cauda), while approximately 17% were exclusive to a single segment, and the remaining percentage was identified in 2 of 3 segments interrogated [57].

#### 4.2.2. Transfer RNA Derived Small RNAs

The most notable differences between the caput and cauda sperm sRNA repertoire are changes in miRNA and tsRNA abundance. Caput sperm display higher proportions of miRNAs and cauda sperm display higher proportions of tsRNAs. Interestingly, these observations are mirrored in caput and cauda epididymosomes, respectively [44]. A report from 2016 provided evidence that epididymosomes were in fact capable of directly delivering sRNAs to sperm cells and that this is the primary mechanism responsible for remodeling of sperm sRNA payloads during epididymal transit. This report also identified miRNAs and tsRNAs which exist at high abundance in epididymosomes, including previously reported let-7 species, miR-34c, tsRNA-Gly-GCC and tsRNA-Val-CAC [44].

#### 4.2.3. Functions of Epididymosomal sRNAs

Considering the significant contribution that epididymosomal protein cargo has to sperm competency it is possible that epididymosomal sRNA content has equally important roles. Various potential functions have been hypothesized in recent years including communication between non-adjacent epithelial cells, support of embryogenesis and epigenetic inheritance. Exploration into these areas is still ongoing and is at the forefront of male reproductive research.

#### 4.2.4. Epithelial Cell Communication

A primary, known function of exosomes is mediating cell-to-cell communication. This process has been extensively studied in some areas of research, such as oncology [59]. It has been hypothesized that epididymosomes may function similarly and serve as a mechanism for communication between non-adjacent cells of the epididymal epithelium or between regions of the epididymal epithelium [58,60]. MicroRNA cargo in epididymosomes isolated from a particular segment of the epididymis partially mirrors miRNA expression patterns in epithelial cells from that same segment. However, some miRNA signatures differ between epididymosomes and their associated epithelial cells, suggesting that miRNA sorting into epididymosomes may be a selective process. Evidence has shown that epididymosomes are capable of binding to epithelial cells in vitro [58]. It is, therefore, possible that delivery of sRNA cargo to epithelial cells by epididymosomes in vivo may functionally regulate gene expression patterns in these cells, however, significantly more research is needed in this area. An inducible, caput-specific knockout mouse model has been established and has been used to explore the effect of caput epididymis specific gene knockout on sperm maturation and fertility [61,62,63]. This model could also be an interesting method for interrogating epithelial cell-to-cell communication between regions of the epididymis.

#### 4.2.5. Support of Embryogenesis

Increasing evidence suggests that the contribution that sperm makes to the developing embryo is greater than just providing genomic DNA. Sperm epigenetics has become a highly studied area of research that includes potential functions of sRNAs contained in sperm. Considering that epididymosomes are largely responsible for establishing the sRNA payloads observed in mature sperm, sRNA cargo in epididymosomes has been interrogated in regard to these functions.

The previously mentioned study by Conine et al. demonstrates striking results of embryonic failures in embryos generated using ICSI from immature, caput sperm as compared to embryos generated from either testicular or cauda sperm. Additionally, this group noted altered gene expression at preimplantation stages in caput-derived embryos [43]. In a parallel publication, this group described sRNAs which are delivered by epididymosomes and therefore exist at altered abundances in testicular, caput and cauda sperm. Specific sRNA species, primarily miRNAs, which can be identified in testicular and cauda sperm but at a significantly reduced abundance, if at all, in caput sperm were implicated in these developmental defects. All of the implicated sRNAs can initially be observed in testicular sperm but are seemingly lost and subsequently replenished by epididymosomes during epididymal transit [44,45]. This group attempted to rescue their caput embryos by microinjecting sRNAs isolated from cauda epididymosomes into caput embryos and observed significant increases in embryo survival rates and normal preimplantation gene expression. These data suggest that sRNAs delivered by epididymosomes are required for normal embryogenesis [43].

Contrasting observations have been made, however, where offspring could be successfully generated using ICSI with caput sperm [46,47]. Multiple explanations for these conflicting results have been proposed, including minor differences in methods and the use of different mouse strains [43]. Additionally, the rationale behind testicular sperm losing competency upon reaching the caput and subsequently gaining it back during epididymal transit remains elusive. Further exploration is needed to determine why and how this loss of miRNAs is occurring, and how consistent it may be among individuals and species. These conflicting results demonstrate that the contribution of epididymosomal and sperm sRNAs to embryogenesis is still very unclear.

Research has also been conducted to interrogate specific sRNAs that are delivered by epididymosomes, such as miR-34c. A study from 2012 microinjected a miR-34c inhibitor into zygotes and observed a significant increase in failure of these embryos to undergo the first cleavage, as compared to embryos injected with a scrambled inhibitor [64]. A later study attempted to recapitulate the results from this 2012 study. Some discrepancies were observed between the results of the two studies. Specifically, the later study by Yuan et al. accounted for miR-34c being part of two clusters of miRNAs all containing the same seed sequence. This group used knock-out (KO) mice that do not express miR-34b/c (one cluster) or miR-449a/b/c, an additional cluster with the same seed sequence. This group did not observe failed cleavage in embryos generated from single KO males, however, they did observe preimplantation development defects in ICSI embryos from double-KO sperm, among other phenotypes [65]. It is likely since the 2012 study did not consider miRNAs with a similar seed sequence, that their inhibitor had off-target effects which would also inhibit miR-34c clusters and not miR-34c alone. This would explain why the two studies, while not being in complete agreement, reported similar embryonic defects. In a 2014 study assessing reprogramming and progression of bovine somatic cell nuclear transfer (SCNT) embryos, an inducible miR-34c expression system was used. Interestingly, doxycycline induction of miR-34c in donor cells used for preparing SCNT embryos resulted in increased cleavage rates in these embryos and altered embryo quality [66]. Predictably, miR-34b and miR-34c were also implicated in the results seen by Conine et al. [43].

While it has been extensively studied, miR-34c is not the only epididymosomal sRNA that may play a role in embryogenesis. In a study of sperm from 102 male patients undergoing ART, sRNA sequencing and subsequent differential expression analysis revealed that high levels of miR-191-5p in sperm were significantly associated with increased fertilization rates (FR), high quality embryo rates (HQER) and effective embryo rates (EER). In the mouse, miR-191-5p is observed at high abundance in caput epididymosomes. Interestingly, and in contrast to data reported in the mouse, this particular study did not observe an association between miR-34c levels in sperm and either FR, HQER or EER [67]. However, to our knowledge, combined expression levels of miR-34c clustered miRNAs were not assessed.

Beyond miRNAs, tsRNAs are also observed at a high abundance in epididymosomes and are delivered to sperm during epididymal transit. Multiple studies have presented evidence that tsRNA-Gly-GCC is one of the most abundant tsRNAs in epididymosomes and is delivered to sperm during epididymal transit [44,68]. In one publication, authors also conducted inhibitory experiments using microinjection of an antisense oligonucleotide to interfere with tsRNA-Gly-GCC and reported altered preimplantation gene expression patterns. Developmental progression was only assessed until the 4-cell stage with no apparent alterations, however, these data do suggest that tsRNAs delivered by epididymosomes to sperm may functionally regulate gene expression in the preimplantation embryo [44]. A recent publication suggests that tsRNA-Gly-GCC mediates processing of histone mRNAs thereby altering chromatin structure and subsequent preimplantation gene expression [69].

Further mechanistic studies are required to better understand the roles of epididymosomal sRNAs. The current body of literature largely focuses on associations and there are a number of discrepancies between studies. Increased knowledge regarding the mechanisms by which epidymosomal sRNAs may support or alter embryo development will allow for stronger study design and more consistency in results moving forward.

#### 4.2.6. Epigenetic Inheritance

Another major focus of studies on epididymosomal sRNAs is their potential ability to serve as a mechanism of epigenetic inheritance from fathers to offspring. Levels of sRNAs in epididymosomes and sperm have been shown to vary based on many lifestyle and environmental conditions, most notably altered diet and stress [44,68,70,71,72,73,74,75,76,77]. Specifically, tsRNAs and miRNAs have been highly implicated in this process.

A large base of evidence suggests that offspring of males consuming specifically altered diets show increased incidences of metabolic disorders [78,79,80]. Additionally, many studies have shown that pre-conception paternal stress can have an impact on offspring neuro- and behavioral development [81,82,83,84,85]. Concomitantly, increasing evidence suggests that levels of sRNAs in sperm, specifically tsRNAs and miRNAs vary based on paternal diet, stress exposure and exercise [44,68,70,71,72,73,75,77,84,86]. In some cases, it has been shown that these changes in abundance are occurring during epididymal transit, suggesting that epididymosomes mediate these observations.

#### 4.2.7. Inheritance of a Metabolic Phenotype

Zygotic microinjection of sRNAs isolated from the sperm of altered diet fed mice has been the preferred method for studying the transmission of metabolic phenotypes to offspring. Evidence suggests that a primary pathway for sRNA biogenesis in sperm occurs via delivery by epididymosomes [44,45]. Therefore, while many studies have focused on interrogating sRNAs in sperm it is likely that the observed alterations in abundance are caused by changes in epididymosome cargo or delivery.

Some authors have reported the ability to regulate offspring metabolic phenotypes using microinjection of sRNAs isolated from sperm. One group fed male mice either a high fat diet (HFD) or control diet (CD), then used ICSI to generate embryos with either HFD or CD sperm. Offspring from HFD males displayed metabolic phenotypes such as insulin resistance and glucose intolerance as compared to CD offspring. This group then isolated sRNAs from HFD and CD sperm, and injected sRNA preparations into control zygotes. Interestingly, offspring from embryos injected with HFD sRNAs displayed some of the same metabolic phenotypes as un-injected HFD offspring, while offspring from embryos injected with CD sRNAs showed normal metabolic function. To differentiate between sRNA species that may be responsible, this group repeated their experiments but selected their sRNA preparations by size. They discovered that injection of only the 30–40 nucleotide fraction of sRNAs mirrored the results from injection of total sRNAs, suggesting that sRNA species of this size are most important in regulating the metabolic phenotype. MicroRNAs are generally smaller in size, and this 30-40 nt fraction represents most tsRNAs contained in sperm [68].

In a later publication, this same group aimed to identify a mechanistic basis for these observations. With this purpose, they turned to interrogating modifications such as methylation that exist on mature tRNAs and remain on some tsRNAs. They found that some modifications were enriched in tsRNAs isolated from HFD sperm and hypothesized that these modifications may be the drivers of regulating an offspring metabolic phenotype. DNMT2 is the primary methyltransferase responsible for the specific modifications observed and, interestingly, offspring of HFD *Dnmt2^−/−^* males do not display the same metabolic phenotypes as HFD *Dnmt2*^−+/+^ males. Further, offspring from embryos injected with *Dnmt2^−+/+^* HFD sperm tsRNAs displayed metabolic phenotypes, while those offspring from embryos injected with HFD *Dnmt2*^−/−^ sperm tsRNAs did not. Taken together, this evidence suggests that DNMT2 may regulate inheritance of a metabolic phenotype through modification of tsRNAs [77].

Many additional correlative studies also exist, that demonstrate metabolic phenotypes in offspring of high fat or low protein diet fed males, and also demonstrate altered sRNA content in sperm. Additionally, paternal exercise and corrected diet has also been shown to reduce both the offspring metabolic phenotypes and alterations in sperm sRNA signatures [75,84,86]. Some studies have also explored additional aspects of offspring health, beyond inheritance of a metabolic condition. Interestingly, it has been reported that female offspring of males being fed a HFD display reduced reproductive potential, and it has been suggested that this observation may be mediated in part by altered sperm sRNA signatures [87]. Steps are being made to move away from primarily correlative results and further explore the mechanisms through which sRNAs in sperm may mediate epigenetic inheritance of metabolic disorders from fathers [77]. It is also important that studies regarding specific sperm sRNAs identify the origin of these sRNA species to more confidently determine whether they were acquired during epididymal transit or are remnants from spermatogenesis. Finally, additional insight is needed, not only into a mechanism of transmission, but also on a broader scale to identify how and why epididymosomes are delivering altered abundances of sRNAs to sperm under different dietary conditions.

#### 4.2.8. Inheritance of Behavioral Phenotypes

From epidemiologic evidence to studies in animal models, there is a large basis of support for observations that pre-conception paternal stress can impact offspring neuro- and behavioral development. One group identified a subset of miRNAs (miR-29c, miR-30a, miR-30c, miR-32, miR-193-5p, miR-204, miR-375, miR-532-3p and miR-698) in sperm that were associated with stress in male mice. Concurrently, they observed reduced hypothalamic-pituitary-adrenal (HPA) stress axis reactivity in offspring of the stressed male mice. This group was able to induce their offspring phenotype in control animals as well, using microinjection of their subset of miRNAs [74]. Interestingly, a study published in 2017 interrogated whether paternal exercise was capable of rescuing hypothalamic-pituitary-adrenal axis dysregulation in offspring from stressed fathers. They used a wheel-running mouse model and observed a reduction in behavioral phenotypes in female, but not male offspring. These authors also identified a subset of sperm miRNAs (miR-19b, miR-455 and miR-133a) and tsRNAs (tsRNA-Gly and tsRNA-Pro) which showed altered abundance in sperm of exercising mice and suggest that these sRNA species may be, at least partially, responsible for the behavioral phenotypes observed [75].

While neither of these two studies specifically mention epididymosomes, of the 12 miRNAs and 2 tsRNAs interrogated between the two studies, 10 of the miRNAs and both of the tsRNAs have previously been identified in mouse epididymosomes [3]. This suggests the possibility that alterations in sperm sRNA content from stressed fathers are mediated by epididymosomes. In fact, a 2020 publication provides support for this hypothesis. In this study, mouse epididymal epithelial cells were cultured in vitro and treated with the stress hormone corticosterone. Extracellular vesicles (EVs) secreted by either stress-treated or control cells were isolated and subsequently incubated with caput sperm from control male mice. Sperm from these preparations were then used to generate offspring, and stress response in adult F1 offspring was assessed. Surprisingly, offspring generated from sperm incubated with the stress-treated EVs displayed the same hypothalamic-pituitary-adrenal axis dysregulation as had been observed in previous studies [70]. This evidence suggests that vesicles secreted from epididymal epithelial cells, such as epididymosomes, are capable of mediating epigenetic inheritance of neuro- and behavioral disorders from fathers.

## 5. Conclusions and Future Directions

Epididymosomes are still a relatively new discovery within the epididymis, an essential reproductive organ with well characterized functions. Incredible progress has been made, especially in the last decade, at determining what role these epididymosomes and their cargo may play in reproduction. However, evidence for these roles at this stage still tends to be inconsistent, with many groups reporting slightly, to completely different results despite apparently strong study design and techniques. Additionally, a massive gap in knowledge exists both up- and downstream of the altered preimplantation gene expression observed in association with epididymal sRNAs. Upstream, little is known about how sRNAs (especially tsRNAs) are functioning within the embryo. It has been suggested that they post-transcriptionally regulate gene expression, however, the mechanism of action and specificity of targets is largely unknown. Downstream, it is unknown how altered preimplantation gene expression translates into the developmental, metabolic, or behavioral alterations observed in embryos and offspring. In many respects, these knowledge gaps are unsurprising considering there is generally limited understanding of the molecular events that govern genome activation and other critical pathways in early development. Deliberate interrogation of intricacies in genome activation and transcription factor signaling cascades might provide much needed clarity as to how epididymosomal sRNA cargo could direct larger changes in embryos and offspring.

In the study of the epididymis, historical research provides interesting perspectives and insights into current literature. Advances in assisted reproductive techniques (ART) have allowed clinicians and researchers to bypass some limitations native to male reproduction that exist within the epididymis such as a necessity for motile and mature spermatozoa. Some evidence does suggest, however, that ART may come with a health cost to offspring [88]. Increased knowledge into potentially novel functions of a historically well described organ such as the epididymis may allow for new treatment options, not only for natural conception but also to increase positive health outcomes in children conceived with ART.

Epididymal and specifically epididymosomal function is clearly an increasingly popular area of research and many high-quality studies have been and continue to be published on the subject. Intriguingly, research in this area spans a broad spectrum, including basic science principals yet having direct translational research implications as well. Current research, as detailed in this review, suggests that clinical consequences of epididymosomes and their cargo are far-reaching. Epididymosome-mediated effects span many fields, including those that are directly related to public health such as metabolic disease, behavioral disorders and infertility. Taken together, this review suggests that an increasing body of evidence is working to develop a complete picture of the role of the epididymis in male reproduction and beyond to offspring health and disease susceptibility.

## Figures and Tables

**Figure 1 ijms-21-05377-f001:**
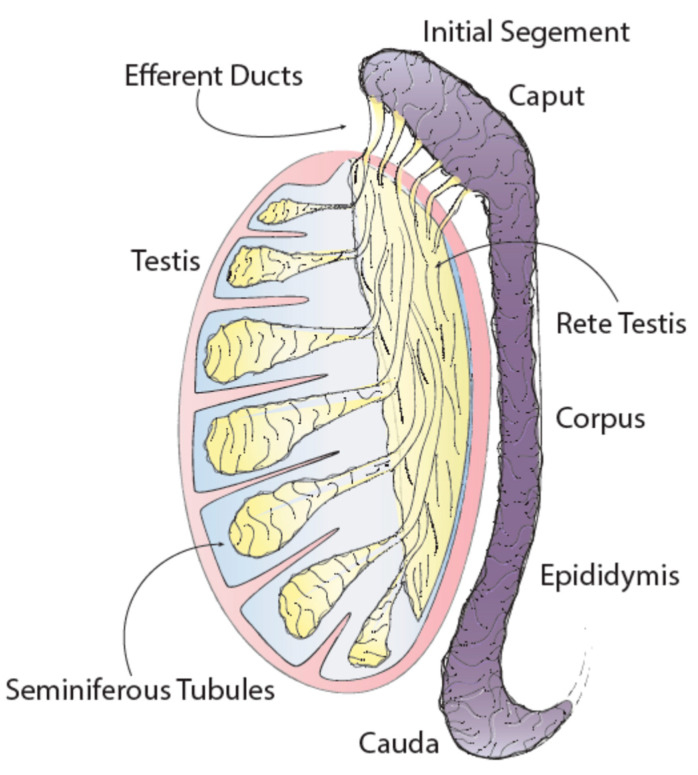
Testis and epididymis anatomy.

**Figure 2 ijms-21-05377-f002:**
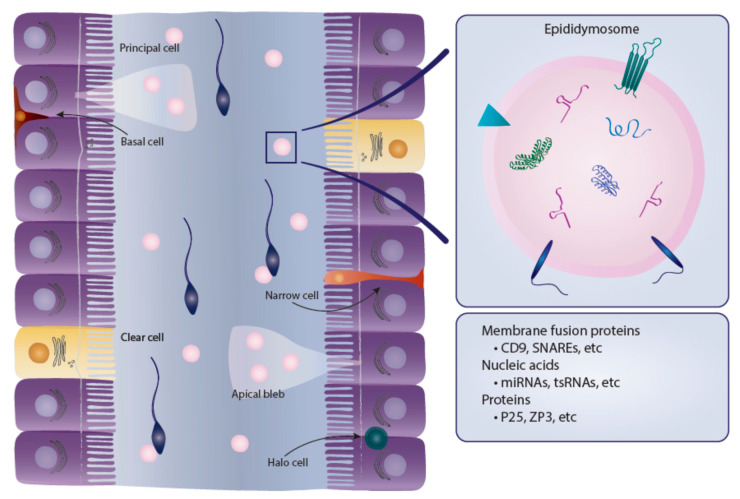
Epididymal internal structure and epididymosomes content.

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
