# Peer review of "The Role of the Epididymis and the Contribution of Epididymosomes to Mammalian Reproduction"

_ijms, 2020, doi:10.3390/ijms21155377_

Round 1
Reviewer 1 Report
I have read the review with great interest. The topic, the contribution of epididymosomes to mammalian reproduction is relevant and attracting more and more attention. The manuscript is well written and although some recent reviews have been published addressing in part the same topic. The design of the present review differs from the recently published reviews and therefore overlap is limited.
I have two minor comments:
- L.67-68 Authors give here the impression that sperm cells directly develop from PGCs. That is of course not the case. These PCGs first have to differentiate into spermatogonial stem cells rom which then A spermatogonia can develop and sperm.
- L.389-393 it may be good to discuss in this respect also the use of testicular sperm in ICSI, sperm that has not been in contact with epididymosomes, and embryo quality/implantation.
Author Response
July 2020
Mr. Jeffery Zhu
Assistant Editor of International Journal of Molecular Sciences
Dear Mr. Jeffery Zhu
Enclosed please find the revised manuscript No. ijms-875334 entitled “The role of the epididymis and the contribution of epididymosomes to mammalian reproduction” which we would like to be reconsidered for publication in International Journal of Molecular Sciences.
We sincerely thank the Editors and Reviewers for the opportunity to have our manuscript reviewed in International Journal of Molecular Sciences, for the global appreciation of the submitted manuscript, and for the valuable and constructive comments on the first version of the manuscript. We have tried to address in detail and accordingly all of the concerns and questions from the editors and reviewers. We provide itemized responses to each point raised by each of the Reviewers and included the changes (track changes mode in MS Word) and comments in the manuscript, when required.
-----------------------------------------------------------------------------------------
Reviewer #1:
I have read the review with great interest. The topic, the contribution of epididymosomes to mammalian reproduction is relevant and attracting more and more attention. The manuscript is well written and although some recent reviews have been published addressing in part the same topic. The design of the present review differs from the recently published reviews and therefore overlap is limited.
We sincerely thank Reviewer #1 for the global appreciation for our manuscript, as well as for all the valuable comments and suggestions provided in the following lines, which have greatly improved the first version of the manuscript. We have addressed all them in each of the following points, as well as in the manuscript, when required. Please find below the itemized responses to the Reviewer #1’s comments.
I have two minor comments:
- L.67-68 Authors give here the impression that sperm cells directly develop from PGCs. That is of course not the case. These PCGs first have to differentiate into spermatogonial stem cells from which then A spermatogonia can develop and sperm.
We completely agree with this comment. We have addressed this in lines 67-68 to clarify the earliest steps of germ cell development.
- L.389-393 it may be good to discuss in this respect also the use of testicular sperm in ICSI, sperm that has not been in contact with epididymosomes, and embryo quality/implantation.
As suggested by the reviewer, we have included the information that the experiments in Conine et al. also included testicular sperm (lines 391-393). Additionally, we have added further questions regarding the competence of testicular sperm and supposed incompetence of caput sperm in lines 407-410.
-----------------------------------------------------------------------------------------
We do hope that all the modifications we have made to the manuscript will make it possible for the paper to be accepted and published in International Journal of Molecular Sciences.
We are looking forward to receiving the editorial decision concerning the revised article.
Yours sincerely,
Dr. Albert Salas-Huetos
Address: Andrology and IVF Laboratory, Division of Urology, Department of Surgery, University of Utah School of Medicine, 84180 Salt Lake City, UT,USA.
Contact: +1 (385) 210-5534; E-mail: albert.salas@utah.edu
ORCiD: 0000-0001-5914-6862
Reviewer 2 Report
In my opinion, this I is an excellent review on the epididymis with a focus on epididymosomes. The authors have provided a comprehensive update on the state of the literature on this subject.
I was surprised that there was not inclusion of specific discussion on the role of reproductive hormones on the function of the epididymis with respect to the contribution of epididymosomes to this function. Even if there is little to no research on hormonal actions and epididymosomes, mention of this as an area requiring further investigation would improve the manuscript. I would like to see this addressed in any revisions of the manuscript.
My only other comment pertains to the level of critical interpretation of the literature in this review. The review is an excellent descriptive catalogue of the state of research on this topic but I feel it could be improved with a slightly more cogent rationale for why research is required in certain areas, rather than just saying that more research is needed. There have been attempts to do this, and the Conclusion and Future Directions handle this reasonably well, but it could be improved fruther. For example, with respect to epididymosomes, it is stated that a majority of mechanistic questions remain unanswered but without critical evaluation of what the key and most important questions are, and with argument as to why they need to be answered. Furthermore, this could be extended to explain how this knowledge could be translated to clinical practice.
Author Response
July 2020
Mr. Jeffery Zhu
Assistant Editor of International Journal of Molecular Sciences
Dear Mr. Jeffery Zhu
Enclosed please find the revised manuscript No. ijms-875334 entitled “The role of the epididymis and the contribution of epididymosomes to mammalian reproduction” which we would like to be reconsidered for publication in International Journal of Molecular Sciences.
We sincerely thank the Editors and Reviewers for the opportunity to have our manuscript reviewed in International Journal of Molecular Sciences, for the global appreciation of the submitted manuscript, and for the valuable and constructive comments on the first version of the manuscript. We have tried to address in detail and accordingly all of the concerns and questions from the editors and reviewers. We provide itemized responses to each point raised by each of the Reviewers and included the changes (track changes mode in MS Word) and comments in the manuscript, when required.
-----------------------------------------------------------------------------------------
Reviewer #2:
In my opinion, this I is an excellent review on the epididymis with a focus on epididymosomes. The authors have provided a comprehensive update on the state of the literature on this subject.
We sincerely thank Reviewer #2 for the global appreciation for our manuscript, as well as for all the valuable comments and suggestions provided in the following lines, which have greatly improved the first version of the manuscript. We have addressed all them in each of the following points, as well as in the manuscript, when required. Please find below the itemized responses to the Reviewer #2’s comments.
I was surprised that there was not inclusion of specific discussion on the role of reproductive hormones on the function of the epididymis with respect to the contribution of epididymosomes to this function. Even if there is little to no research on hormonal actions and epididymosomes, mention of this as an area requiring further investigation would improve the manuscript. I would like to see this addressed in any revisions of the manuscript.
We thank the reviewer for this comment. We have included a brief statement regarding hormonal regulation of the epididymis on lines 133-136. Additionally, we have included some information about an androgen-regulated epididymosomal protein on lines 303-306. We have also included a call for additional research into potential functions of hormones in the regulation of epididymosomal release or cargo on lines 320-327. We feel this is an important and relevant topic which was overlooked in the first version of our manuscript and appreciate the reviewer bringing it to our attention.
My only other comment pertains to the level of critical interpretation of the literature in this review. The review is an excellent descriptive catalogue of the state of research on this topic but I feel it could be improved with a slightly more cogent rationale for why research is required in certain areas, rather than just saying that more research is needed. There have been attempts to do this, and the Conclusion and Future Directions handle this reasonably well, but it could be improved further. For example, with respect to epididymosomes, it is stated that a majority of mechanistic questions remain unanswered but without critical evaluation of what the key and most important questions are, and with argument as to why they need to be answered. Furthermore, this could be extended to explain how this knowledge could be translated to clinical practice.
We appreciate the level of detail and criticism from the reviewer regarding this aspect of our manuscript. To address this, we have included a few more specifics regarding areas requiring more research throughout the manuscript (lines 322-327, 407-410). Additionally, we have bolstered the Conclusions and Future Directions section to include better specifics on necessary future work and discussion on clinical fields where this research could be applied (lines 547-558, 575-578).
-----------------------------------------------------------------------------------------
We do hope that all the modifications we have made to the manuscript will make it possible for the paper to be accepted and published in International Journal of Molecular Sciences.
We are looking forward to receiving the editorial decision concerning the revised article.
Yours sincerely,
Dr. Albert Salas-Huetos
Address: Andrology and IVF Laboratory, Division of Urology, Department of Surgery, University of Utah School of Medicine, 84180 Salt Lake City, UT,USA.
Contact: +1 (385) 210-5534; E-mail: albert.salas@utah.edu
ORCiD: 0000-0001-5914-6862